# A visualization system based on Wikidata for supporting and monitoring the Wiki Loves Monuments Italian contest

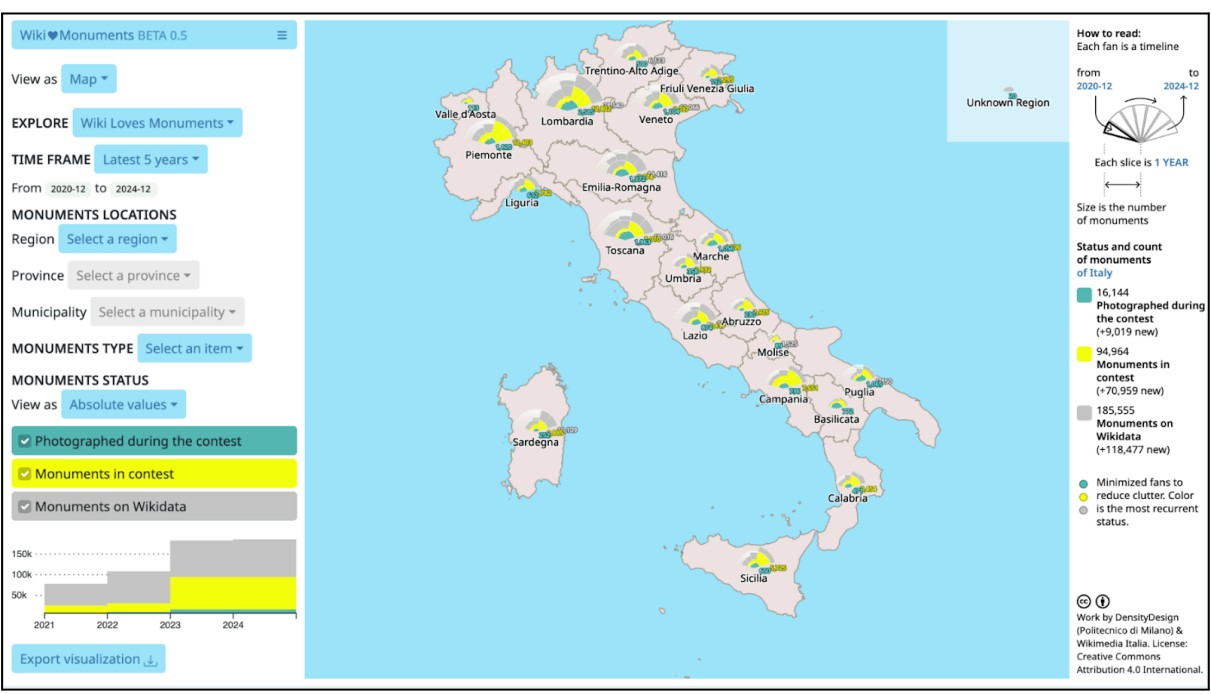

Screenshot of the Wiki Loves Monument Observatory accessible at https://data.wikilovesmonuments.it/

This paper introduces the Wiki Loves Monuments Observatory, an interactive information and visualization system designed to leverage Wikidata (Vrandečić & Krötzsch, 2014) in support of the contest Wiki Loves Monument Italy (WLM Italy). Since 2012, WLM Italy (Azizifard et al., 2023; Bertacchini & Pensa, 2023) has been organized by the local chapter of Wikimedia to enhance the documentation of Italian cultural heritage using content curated in Wikidata and Wikimedia Commons. The observatory facilitates the work of organizers, volunteers, and participants, and helps increase the quality of the materials produced. Specifically, it provides greater support in the following areas: (1) the organization of the national contest as well as regional and local competitions; (2) the engagement of volunteers, both to enhance the database and to produce new photographs; (3) the promotion of the contest through social media and other communication channels; (4) the monitoring of the coverage of Italian cultural heritage documentation on Wikidata and Wikimedia Commons.
The platform consists of a database (DB) and a user interface (UI). The UI (https://data.wikilovesmonuments.it/) empowers volunteers, heritage professionals, and local organizers to plan outreach campaigns, coordinate activities (e.g., wiki-expeditions), and enrich digital cultural repositories leveraging a map visualization and a filterable list of cultural properties. It offers both aggregated and granular views, enabling to understand coverage patterns, identify areas in need of documentation, reveal temporal trends in community-driven efforts, and produce visual reports. The DB (https://wlm-it-visual.wmcloud.org/api/schema/swagger-ui/) is automatically updated using specific Wikidata Sparql queries and API requests to Wikimedia

Commons. It is accessible via unrestricted APIs and currently used by the presented observatory and other projects.

The relevance of this work lies not only in the technical demonstration of using Wikidata as a foundational infrastructure, but also in the participatory design approach (Dörk et al., 2020; Morelli et al., 2021; Sanders & Stappers, 2008) that underpins its implementation. The richness of Wikidata content and the heterogeneity of WLM stakeholders enable virtually limitless possibilities for data analysis. Therefore, identifying and prioritizing the features to be implemented is a complex and non-trivial task that requires activities designed to define, with the final users, what needs to be made available and how. The design methodology entailed desk research, participant observation and structured interviews, which are conducted to inform the realization of a participatory workshop. This step was crucial, as it allowed for the early identification of all necessary features, facilitating the design of a robust, modular, and upgradable infrastructure capable of adapting to the future needs. The resulting system is not merely a dashboard visualization, but a flexible framework where community contributions, iterative refinement, and open data principles converge.

By synthesizing these technical and participatory insights, the contribution offers a replicable model for future efforts to leverage Wikidata in projects related to cultural heritage and volunteer engagement. On one hand, it provides examples of data integration and curation strategies; on the other, it exposes the need for participatory design methods to accommodate and anticipate multiple stakeholders' needs and perspectives.

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
