# OpenReview forum: "A visualization system based on Wikidata for supporting and monitoring the Wiki Loves Monuments Italian contest"
_wikimedia.it/Wikidata_and_Research/2025/Conference — WD&R Paper_

### Official Review · ~Luca_Martinelli1 · 2025-01-04
**Approvo**

**Originality:** 4
**Impact:** 4
**Confidence:** 5

**Review:**

Wiki Loves Monuments è stato il progetto di punta dell'associazione Wikimedia Italia per gli ultimi 15 anni e Wikimedia Italia è stata fra le prime a trasferire il proprio database di codici e autorizzazioni su Wikidata, proprio per sfruttare le potenzialità del progetto nell'organizzazione del contest. Questo tool è un'applicazione che permette di valutare gli sforzi di 15 anni di raccolta di dati, autorizzazioni e immagini riguardanti i monumenti e i luoghi di interesse culturale dell'Italia, ed è pertanto ampiamente coerente con quanto si intende dimostrare sulle potenzialità di Wikidata.

**Compliance:**

5

**Final Paper Review:**

Confermo il mio giudizio positivo già espresso. Si tratta di un interessante excursus su come è stato concepito il tool di visualizzazione dei dati, con importanti annotazioni riguardo la necessità di partecipazione della comunità nello sviluppo di ulteriori tool a servizio della stessa. Decisamente un lavoro valido e da mostrare alla conferenza.

**Scientific Quality:**

5

---

### Official Review · ~Federico_Morando1 · 2025-01-19
**Relevant and outstanding**

**Originality:** 4
**Impact:** 5
**Confidence:** 5

**Review:**

The abstract is outstanding in terms of organization and clarity. It effectively presents multiple angles for assessing the paper's contribution (and the related project), such as the impact of the developed software and the methodology used in its design. Additionally, the analysis of the collected data provides another important layer—arguably the most significant one.

Naturally, assessing the originality of the work in terms of scientific contribution is challenging based solely on such a brief abstract. However, this submission stands out, in my opinion, should definitely be accepted.

**Compliance:**

5

**Final Paper Review:**

I fully confirm my previous positive assessment regarding the paper. However, while the assertion that

"The implementation of the presented tool, released in 2022, serves as the foundation for the development of a companion application dedicated to volunteers. Built on the same infrastructure, the new application introduces features aimed at facilitating user contributions rather than monitoring. It yields strong quantitative results, making the Italian contest the edition with the highest number of submitted contributions."

may indeed be accurate, the causal relationship between the introduction of the tool(s) and the high number of submitted contributions warrants further investigation. Specifically, it would be valuable to perform a simple comparative analysis of the Italian WLM contest's performance over time relative to other countries, to better understand whether the observed increase is attributable to the new tool or part of a broader trend.

**Scientific Quality:**

5

---

### Decision · Program_Chairs · 2025-02-05

Accept (Paper)